# Is Exploration or Optimization the Problem for Deep Reinforcement Learning?

## Abstract

In the era of deep reinforcement learning, making progress is more complex, as the collected experience must be compressed into a deep model for future exploitation and sampling. Many papers have shown that training a deep learning policy under the changing state and action distribution leads to sub-optimal performance, or even collapse. This naturally leads to the concern that even if the community creates improved exploration algorithms or reward objectives, will those improvements fall on the *deaf ears* of optimization difficulties. This work proposes a new *practical* sub-optimality estimator to determine optimization limitations of deep reinforcement learning algorithms. Through experiments across environments and RL algorithms, it is shown that the difference between the best experience generated is 2-3$\times$ better than the policies' learned performance. This large difference indicates that deep RL methods only exploit half of the good experience they generate.

## 1 Introduction

What is preventing deep reinforcement learning from solving harder tasks? Many papers have shown that training a deep learning policy under the changing state distribution (non-IID) leads to sub-optimal performance (Nikishin et al., 2022; Lyle et al., 2023; Dohare et al., 2024). However, at a macro scale, it is not completely clear what causes these issues. Do the network and regularization changes from recent work improve exploration or exploitation, and which of these two issues is the larger concern to be addressed to advance deep RL algorithms? For example, better exploration algorithms can be created, but will the higher value experience fall on the *deaf ears* of the deep network optimization difficulties?

How can we understand if the limited deepRL performance is due to a lack of good exploration or deep network optimization (exploitation)? Normally in RL, to understand if there is a limitation, an oracle is needed to understand *sub-optimality*, how far the algorithm is from being optimal. However, that analysis is with respect to the best policy and aliases both causes of the limitations of either exploration or optimization. Instead, consider the example where a person is learning how to build good houses. There are two issues that may prevent the person from *consistently* building a high quality house: (1) they can't *explore* well enough to discover a good design or (2) they can explore well enough to find good designs, but they can't properly *exploit* their experience to replicate those good experience. For deep RL algorithms, which of these two issues is more prevalent?

To understand if exploration or exploitation is the larger culprit, a method is needed to estimate the *practical sub-optimality* between these cases. This estimator should (1) measure the agent's ability to explore, (2) while also estimating the average performance for the learning policy $\hat{\pi}^\theta$. While estimating the average policy performance is common, estimating the exploration ability for a policy is not. Extending the house-building metaphor, the idea is to estimate how close the agent ever got to constructing a good home. Therefore, to realize this estimator, we propose computing the *exploration* value for a policy that is calculated over prior experience, called the *experience optimal policy*. Using this concept, a new version of sub-optimality can be developed that can compute the difference between the *experience optimal policy* and the learned policy, shown in Figure 1a. If there is a large difference between these two, then the performance is limited by exploitation and optimization (model); however, if the difference between the *experience optimal policy* and the *learned policy* is small, then performance is limited by exploration (data).

The described estimator is used to better understand the reason deepRL algorithms do not solve certain *difficult* tasks. It is found that the limitation of deepRL agents in making progress on difficult tasks is not exploration but often exploitation. Therefore, this paper argues that to advance deep reinforcement learning research, further work is needed on optimization for exploitation under non-iid data. The proposed metric can serve multiple additional purposes. (1) For any RL practitioner, this metric can be used to quickly identify if the limitation in performance is an exploitation or exploration problem so that they can focus their efforts. (2) For the research community, this metric can be used across environments and algorithms to understand the performance of deepRL algorithms better and shed light on the *exploration* vs *exploitation* trade-off on a macro sense, to determine if to increase RL progress the community should be working more on exploitation problems[1]. (3) Showing that including exploration bonuses or scaling network size with RL algorithms increases the practical sub-optimality, indicating that optimization becomes a larger issue in that setting. Section 5 provides evidence to showcase these uses of this new view on sub-optimality, and finds for many environments the difference between the *experience optimal policy* and the learned policy to be larger than the difference between the learned policy and the initial policy. These findings suggest a significant exploitation issue and a need for improved optimization methods in RL.

## 2 RELATED WORK

Since the first successes of RL and function approximation (Tesauro et al., 1995; Mnih et al., 2015; OpenAI et al., 2019), many recent works have shown great progress on integrating the complexities of deep learning and reinforcement learning (Hansen et al., 2022; van Hasselt et al., 2018). Many have studied that certain model classes and loss assumptions make it easier to train more performant deepRL policies (Schwarzer et al., 2023; Farebrother et al., 2024), deepRL is even used to fine-tune the largest networks to create strong LLMs (OpenAI, 2022). While deepRL is now being used across a growing number of applications, the broad limitations of current algorithms become less clear.

**Deep Reinforcement Learning Training** The field of methods to explain and improve on the limitations of combining function approximation and reinforcement learning (deepRL) is expanding. Much of the early work consisted of improving value-based methods to overcome training and non-IID data issues in DQN (Mnih et al., 2015) and DDPG (Lillicrap et al., 2015) and stochasticity (Schulman et al., 2015; 2017). Recent adaptations improve over the initial algorithms that struggle with overestimation (van Hasselt et al., 2016; Bellemare et al., 2017; Hessel et al., 2018) or improving critic estimation (Fujimoto et al., 2018; **?**; Lan et al., 2020; Kuznetsov et al., 2020; Chen et al., 2021). The challenges in the space of learning policy are based on an unstable mix of function approximation, bootstrapping, and off-policy learning, called the Deadly Triad in DRL (van Hasselt et al., 2018; Achiam et al., 2019). Many works focus on parts of the triad, including: stabilizing effect of target network (Zhang et al., 2021c; Chen et al., 2022; Piché et al., 2022), difficulty of experience replay (Schaul et al., 2016; Kumar et al., 2020; Ostrovski et al., 2021), over-generalization (Ghiassian et al., 2020; Pan et al., 2021; Yang et al., 2022), representations in DRL (Zhang et al., 2021a; Li et al., 2022; Tang et al., 2022), off-policy correction (Nachum et al., 2019; Zhang et al., 2020a; Lee et al., 2021), interference (Cobbe et al., 2021; Raileanu and Fergus, 2021; Bengio et al., 2020) and architecture choices (Ota et al., 2020).

**DeepRL Exploration Methods** On top of the above training stability improvements is the desire to improve exploration by providing the agent with better signal to encourage exploration beyond just the extrinsic reward. These intrinsic rewards often compute some measure of state visitation or mutual information using a separate online learnt model. Count-based methods (curiosity) are early examples that encourage agents to cover a larger state space (Bellemare et al., 2016; Ostrovski et al., 2017; Tang et al., 2017; Burda et al., 2018a), but they do not scale well to large state spaces. Several works (Pathak et al., 2017; Badia et al., 2020; Zhang et al., 2020b; 2021d) have built on curiosity frameworks to improve training and learning. However, it is not known how well RL algorithms will be able to learn from the additional experience.

---

[1]This is not a judgement on the exploration community, in fact it is with exploration community in mind this work started so that their amazing research gets the best analysis it can, and great exploration algorithms are not misunderstood due to exploitation problems.

**DeepRL Scaling Methods** Given the significant gains of using large models on many supervised learning problems, the RL community has been studying how to achieve similar gains from scale, but deep RL performance often drops when larger networks are used (Schwarzer et al., 2023; Tang and Berseth, 2024). Recent works focus on network structure changes to avoid divergence and collapse, using normalization layers (Nauman et al., 2024; Lyle et al., 2024), regularization (Nikishin et al., 2022; Schwarzer et al., 2023; Galashov et al., 2024) or optimization adjustments (Lyle et al., 2024). The goal in these prior works is to understand and improve performance when larger networks are used, but these papers are often limited to recovering prior performance, not understanding where RL in general is missing potential.

**Convergence and Exploration Theory** The challenges of reinforcement learning algorithms in finding optimal policies are not a new question. Many prior works have studied the theoretical implications on convergence rates (Bhatt et al., 2019; Agarwal et al., 2020; Zhang et al., 2021b; Bhandari and Russo, 2024; Montenegro et al., 2025); however, these studies are limited to linear and tabular models and can not provide a wider lens on the challenges of convergence analysis in the case of deep RL with large function approximators and beyond just policy gradient analysis. A related question is how optimization can be made more robust with regularizers such as entropy (Ahmed et al., 2019; Husain et al., 2021). A recent work in this area studies how intrinsic rewards can influence and improve policy convergence (Bolland et al., 2025). However, those results are for simple environments with limited experimental analysis.

## 3 BACKGROUND

In this section, a very brief review of the fundamental background of the proposed method is provided. reinforcement learning (RL) is formulated within the framework of an Markov Decision Processes (MDP) where at every time step $t$, the world (including the agent) exists in a state $\mathbf{s}_t \in \mathcal{S}$, where the agent is able to perform actions $\mathbf{a}_t \in \mathcal{A}$. The action to take is determined according to a policy $\pi(\mathbf{a}_t|\mathbf{s}_t)$ which results in a new state $\mathbf{s}_{t+1} \in \mathcal{S}$ and reward $r_t = R(\mathbf{s}_t, \mathbf{a}_t)$ according to the transition probability function $P(\mathbf{s}_{t+1}|\mathbf{s}_t, \mathbf{a}_t)$. The policy is optimized to maximize the future discounted reward $\mathbb{E}_{r_0,...,r_T}\left[\sum_{t=0}^{T} \gamma^t r_t\right]$, where $T$ is the max time horizon, and $\gamma$ is the discount factor. The formulation above generalizes to continuous states and actions. There are multiple RL algorithms that can be used to optimize the above objective. This work uses two of the most popular algorithms DQN (Mnih et al., 2015) and PPO (Schulman et al., 2017) to frame the challenges with optimizing and exploration.

**Policy Gradient Definitions** To discuss the difference between policy performance and estimators, it is useful to define the state visitation distribution $d_{s_0}^\pi(s)$ for a policy:

$$d_{s_0}^\pi(s) := (1 - \gamma)\sum_{t=0}^{\infty} \gamma^t \Pr(s_t = s|s_0), \tag{1}$$

where $\Pr^\pi(s_t = s|s_0)$ is the probability of the policy $\pi$ visiting the future state $s_t$ when starting from $s_0$. The policy gradient can be written in the form

$$\nabla_\theta V^{\pi_\theta}(s_0) = \frac{1}{1-\gamma}\mathbb{E}_{s \sim d_{s_0}^{\pi_\theta}}\mathbb{E}_{a \sim \pi_\theta(\cdot|s)}\left[\nabla_\theta \log \pi_\theta(a|s)Q^{\pi_\theta}(s, a)\right]. \tag{2}$$

Then we can write out the **performance difference lemma** (Kakade and Langford, 2002) between two policies as

$$V^{\pi'}(s_0) - V^\pi(s_0) = \frac{1}{1-\gamma}\mathbb{E}_{s \sim d_{s_0}^\pi}\mathbb{E}_{a \sim \pi(\cdot|s)}\left[A^{\pi'}(s, a)\right]. \tag{3}$$

Where $A^{\pi'}(s, a)$ is the advantage of policy $\pi'$.

## 4 IS EXPLORATION OR EXPLOITATION THE ISSUE FOR DEEPRL?

Often, learning agents are concerned with the exploration vs exploitation trade-off and its effect on performance. This trade-off is a helpful lens for discussing an agent's choices at a particular state $\mathbf{s}_t$,

but this single state view focuses on *exploitation* as either: a type of greedy action selection, sampling from a learned policy, or utilizing a world model. However, in the age of deep learning and ever increasing model and data sizes, that lens misses the broader idea that *exploitation is making use of prior experience*, in that for each of the original definitions, there is an imperfect optimization process of the network parameters $\theta$ over some experience $\mathcal{D}$ causing a difference in performance. However, it is often unclear if the difference is from data distribution issues (Ostrovski et al., 2021) or optimization (Lyle et al., 2024). To improve the understanding of the limitations of RL with function approximation (deepRL), we introduce estimators to quantify the difference between a policy's data-generating process (exploration/data) and its ability to learn from that data (exploitation/model).

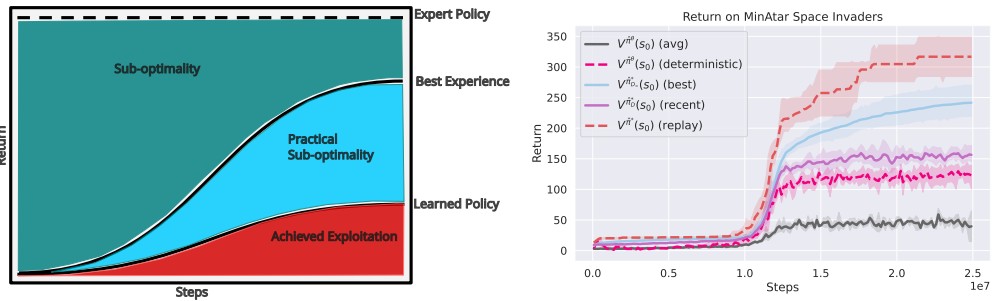

(a) Example exploitation sub-optimality difference

(b) MinAtar space invaders DQN

Figure 1: Left: Diagram of the practical sub-optimality = Best Experience - Learned Policy. On the right are results computing this exploitation gap as the difference between $V^{\hat{\pi}^*}(s_0)$ and $V^{\hat{\pi}^{\theta}}(s_0)$ in MinAtar SpaceInvaders.

In Figure 1 we show the conceptual version of studying this exploration vs exploitation problem, where the typical learning graph is now split into three sections: the performance of the average policy $\hat{\pi}^{\theta}$ from *achieved exploitation* (red), which measures what that policy has learned, the potential performance, indicated by the optimal policy $\pi^*$ (green), and a new estimator we call the *experience optimal* policy $\hat{\pi}^*$ (blue). The challenge is that $\hat{\pi}^{\theta}$ can be arbitrarily bad compared to $\pi^*$, and normally it is not clear if the performance difference (Equation (3)) is because the agent is not exploring well ($V^{\hat{\pi}^*}(s_0) << V^{\pi^*}(s_0)$ and $V^{\hat{\pi}^{\theta}}(s_0) << V^{\pi^*}(s_0)$) or just not exploiting well ($V^{\hat{\pi}^{\theta}}(s_0) << V^{\pi^*}(s_0)$). Understanding if the policy is generating high value trajectories can be particularly useful for evaluating exploration-focused algorithms. When evaluating the performance of a method, if only $V^{\hat{\pi}^{\theta}}(s_0)$ is considered, the analysis can miss the fact that the method is generating higher value experiences $V^{\hat{\pi}^*}(s_0)$, but the policy is not able to exploit them into $\theta$ properly. Therefore, to better understand reinforcement learning limitations, we introduce a new estimator for $\hat{\pi}^*$ to measure practical sub-optimality, which estimates the realizable performance of the policy because the policy has generated behavior with higher value.

**How to measure practical sub-optimality** The optimal policy is defined as the policy that selects the best action at every state (Bellman, 1954). Sub-optimality measures the difference between a policy's value $V^{\pi}(s)$ with respect to an optimal policy $\pi^*$ with the value function $V^{\pi^*}(s)$. However, if the policy $\pi$ struggles to learn from optimal data or explore well, measuring against $\pi^*$ does not tell us if cause of the performance gap is due to exploration of exploitation. Therefore, in addition to the theoretical optimal policy, we introduce the *experience optimal* policy $\hat{\pi}^*$ to represent the best policy the agent can achieve given the experience collected during training. If the environment is deterministic and the agent keeps a buffer of all prior experience $D^{\infty}$, then,

$$\hat{\pi}^* = \underset{<a_0,...,a_t>\in D^{\infty}}{\arg\max} \sum_{t=0}^{T} R(a_t, s_t) \tag{4}$$

This policy can also be understood as deterministically replaying the highest value sequence of actions $<a_0, \ldots, a_t>$ in the experience memory. This policy can be used to compute a new difference as the *practical sub-optimality* of the form $V^{\hat{\pi}^*}(s_0) - V^{\hat{\pi}^{\theta}}(s_0)$.

Most empirical works use the performance of the learned policy $V^{\pi^{\theta}}(s_0)$ to comparing across algorithms to understand which algorithm performs the best on a set of tasks. While this model works

well and enables the community to make steps forward in terms of performance, the learned policy does not provide information on why one algorithm is better than another. Consider the example where there are two algorithms A and B, algorithm A generates higher-value experience, but is not able to exploit them, and B does not generate higher-value experience ($V^{\hat{\pi}_A^*}(s_0) > V^{\hat{\pi}_B^*}(s_0)$), similar to its policy, but has been able to exploit that data well. Both A and B can have the same value $V^{\hat{\pi}_A^\theta}(s_0) = V^{\hat{\pi}_B^\theta}(s_0)$. Is A or B the worse RL algorithm? In this work, we propose that A is the worse algorithm, as it can not properly exploit its generated data. If the experience were equal, algorithm B would see the same experience as algorithm A, then B would result in better performance and have a smaller practical sub-optimality.

## 4.1 Softer Expert Estimators

While Equation (4) is a clear definition for computing an estimate of an optimal policy where, for example, the sequence of actions $a_0, \ldots, a_t$ can be replayed in the environment to reproduce $V^{\hat{\pi}^*}(s_0)$, however, this restrictive definition is subject to high variance and it is less useful for non-deterministic environments. Therefore, two additional methods are introduced to estimate the *potential* for the policy to learn from its experience to indicate that not only is there one high value trajectory, but many higher value trajectories.

For the analysis, two versions of $V^{\hat{\pi}^*}(s_0)$ are introduced to approximate the performance on the *best experience*. For stochastic environments, the first version is the best policy from the collected experience as the top $k\%$ of experience generated by the agent $V^{\hat{\pi}_{D_\infty}^*}(s_0)$, where $D_\infty$ is all the experience collected by the agent. The second is the *recent* top $k\%$ of data $V^{\hat{\pi}_D^*}(s_0)$ in the replay buffer $D$. To estimate the value function $V(s_0)$ from data, the sum of rewards the agent achieves in the environment, or the return, is used. The value estimate is computed using the following function:

$$V^{\hat{\pi}^*}(s_0) = \frac{1}{n} \sum_{\tau \in D_{0:n}} \sum_{a_t, s_t \in \tau} R(a_t, s_t) \tag{5}$$

Where $m$ is equal to $k \times |D|$ and $D$ is sorted with the highest value trajectory starting at index $0$. Our sensitivity analysis in appendix A.1 finds that a value of $5\%$ for $k$ is a good balance over using a single trajectory.

The best *ever* and *recent* estimators both have their own reasoning. The best *ever* experience $V^{\hat{\pi}_{D_\infty}^*}(s_0)$ is a measure of how good the agent is at exploiting the best experience it ever generated. This notion is rather strong and difficult for any *deep* RL algorithm to match, as the agent may not currently have access to that experience for optimization, but it is a notion of lifetime achievement and represents a possible high-value policy and trajectory the agent could generate again. The *recent* best experience $V^{\hat{\pi}_D^*}(s_0)$ is a measure of the agent's ability to learn to match the best of the recent experience it has access to and can use for exploitation. The *recent* notion can be more fair as it is possible for an agent to train on that experience to improve its performance actively, but as will be shown in Section 5.1, RL algorithms also struggle to match this performance.

## 4.2 Comparing RL Algorithms

The above estimators can be used to understand the practical sub-optimality of an algorithm on an environment. That information is useful, but it does not provide information about how an algorithm performs holistically. For example, we may have the question, *how much does an algorithm suffer from exploitation limitations* or *which algorithms are the best at exploiting their generated data*? The easier we can answer the above questions the easier one can focus on making improvents to their algorithm. This information is paramount for the community to understand better where there is a larger benefit from time spent on research and development. To compute this information *across environments* $\mathcal{T}$, the estimator can to be aggregated and normalized across environments.

To compute this aggregate estimator the upper bound from $V^{\hat{\pi}^*}(s_0)$ can be used in place of the *optimalality gap* from *rliable* (Agarwal et al., 2021). The gap computed using the proposed metric is relative to the experience the agent has generated, which can provide more rich signal than comparing the performance to some current notion of a human level agent. For example, in practice when the optimal performance $V^{\pi^*}(s_0)$ is not know a heuristic is used to compute the optimal return for the

expert by taking the max possible reward $r_{\max}$ and multiplying this by the inverse of the discount factor $V^{\pi^*}(s_0) \approx r_{\max} * \frac{1}{1-\gamma}$, which can be far above the optimal policy's performance. To provide a more grounded upper bound on performance by using achieved experience we can instead use the practical sub-optimality:

$$\frac{1}{|\mathcal{T}|} \sum_{m \in \mathcal{T}} (V_m^{\hat{\pi}^*}(s_0) - V_m^{\pi^\theta}(s_0))/(V_m^{\hat{\pi}^*}(s_0) - V_m^{\pi^0}(s_0)). \tag{6}$$

Where $m$ is some task or environment. This metric is used in Section 5.4 to compare the aggregate weaknesses across RL algorithms.

**Implementation Details** It is difficult to compute a general practical sub-optimality for any type of RL algorithm. On-policy algorithms do not keep around histories of recent data for evaluation, and off-policy algorithms don't track returns as they often use Q-functions for learning directly from rewards. To facilitate the tracking of these statistics, we develop a wrapper that can be introduced into the RL algorithm code to track every reward, return, trajectory, and end of episode. This wrapper is also used to compute the *best* $V^{\hat{\pi}^*_{D_\infty}}(s_0)$ and the *recent* $V^{\hat{\pi}^*_D}(s_0)$ estimates during learning.

## 5 EXPERIMENTAL RESULTS

In this section, the ability of practical sub-optimality for diagnosing learning issues is evaluated. This usefulness is determined in multiple ways: (1, Section 5.1) As a metric to determine the limitations of current RL algorithms on specific environments, (2, Section 5.2) how recent methods for exploration or scaling increase or reduce the practical sub-optimality, and (3) The overall limitations of RL algorithms and if more exploration or exploitation is needed to improve performance over difficult/unsolved tasks wrt to scaling in Section 5.3 or in general in Section 5.4.

Four popular RL algorithms are used for evaluation. First PPO (Schulman et al., 2017) is a common on-policy algorithm used for various problems, known for its ease of implementation and use. The other algorithm is DQN (Mnih et al., 2015), which is a popular RL algorithm for environments with discrete actions. PQN (Gallici et al., 2025), which is an adapted version of DQN to learn with increased parallelization. Last is SAC (Haarnoja et al., 2018), which is based on maximum entropy optimization, which can be more robust at finding optimal policies. These algorithms cover the most common use cases for RL.

A selection of evaluation environments is included to cover a diverse range of the RL landscape. This diverse selection is important to understand better the practical sub-optimality there needs to be a difference between the generated data and the final policy's performance. Therefore, we focus on including experimental results on environments that are *difficult*. These difficult environments include using MinAtar (Young and Tian, 2019) and Atari (Bellemare et al., 2015; Aitchison et al., 2022) **SpaceInvaders**, **Asterix**, **LunarLander**, **Montezumas Revenge**, **Craftax**, and the **Atari-Five** (Aitchison et al., 2022). We also include **Walker2d**, **HalfCheeta**, **Humanoid** as continuous action environments that are easier, and as will be shown, have little practical sub-optimality.

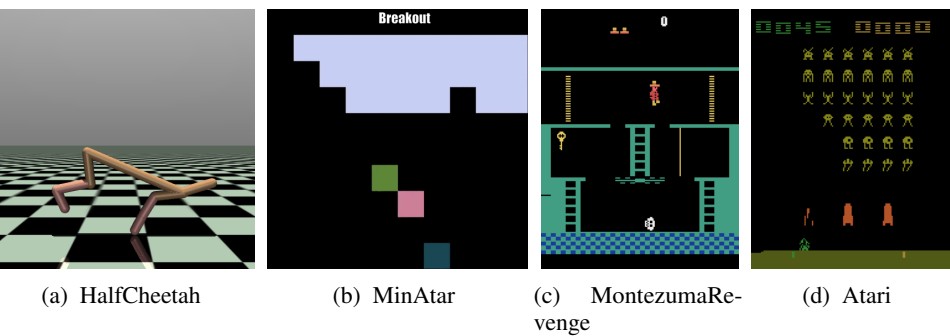

(a) HalfCheetah    (b) MinAtar    (c) MontezumaRe-venge    (d) Atari

Figure 2: Evaluation environments include examples from Mujoco, MinAtar, and Atari.

To measure performance, we will look at the practical sub-optimality discussed in the previous section. In addition, the average return during learning is used to verify that the agents are learning,

ensuring that the reason for the lack of practical sub-optimality is not due to the agent's inability to learn. All experiments are conducted over 4 random seeds.

## 5.1 PER TASK SUB-OPTIMALITY

In this section, we can study which tasks express types of this practical sub-optimality, indicating a need for improvements in optimization over exploration. The first question (1) is whether tasks exhibit this type of gap, or if all tasks can be solved, or if policies can properly exploit the experience. In Figure 3a, we can see that for **HalfCheetah** there is little difference between $V^{\hat{\pi}^*_{D\infty}}(s_0)$, $V^{\hat{\pi}^*_D}(s_0)$, and $V^{\hat{\pi}^\theta}(s_0)$, even though a high return is achieved; however, it is well known that **HalfCheetah** is no longer a difficult task for common RL algorithms. Instead is we look at the **Humanoid** task we can see that even SAC has a gap in performance Figure 8c. We can also see that the deterministic $\hat{\pi}^*$ poorly estimates the best performance in this non-deterministic environment, and instead the softer versions work well[2]. Examining tasks that are well-known to be difficult exploration problems reveals a different story. After training PPO on **Montezuma's Revenge** (Figure 3b), there is a surprisingly large gap where $V^{\hat{\pi}^\theta}(s_0)$ is noisy and near zero, yet the policy does generate many high-value trajectories, indicated by both a large difference between $V^{\hat{\pi}^*_{D\infty}}(s_0)$ and $V^{\hat{\pi}^*_D}(s_0)$, but PPO is not able to learn from these. These higher value trajectories are not rare. The $V^{\hat{\pi}^*_D}(s_0)$ line indicates that, aside from a few spikes, the policy is far from the best $5\%$ of experiences. We find similar results for many other environments and algorithms shown in Figure 3 and for PQN in Figure 11f.

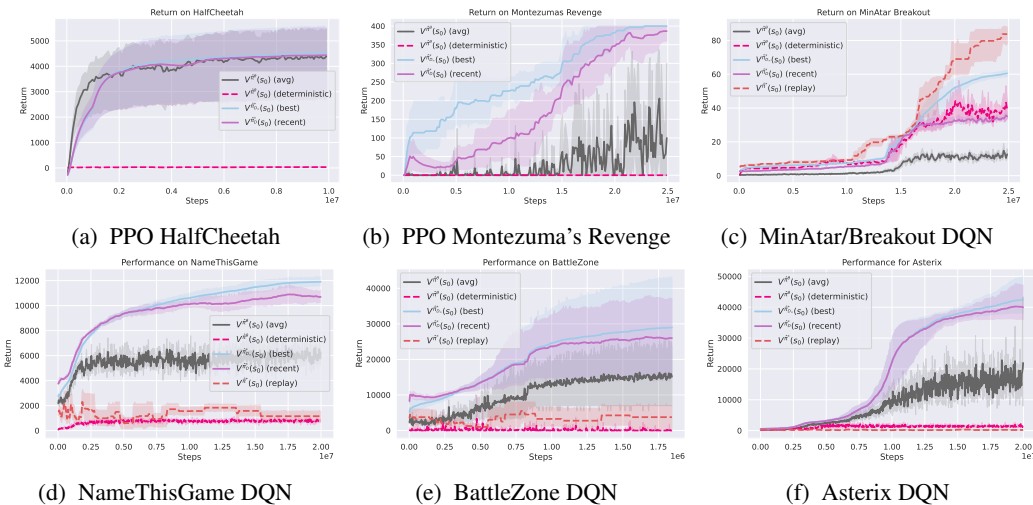

(a) PPO HalfCheetah     (b) PPO Montezuma's Revenge     (c) MinAtar/Breakout DQN

(d) NameThisGame DQN     (e) BattleZone DQN     (f) Asterix DQN

Figure 3: Comparisons of different measures for global optimality and the learned policy $\pi^\theta$. For environments with more complex exploration, such as Montezuma's Revenge, Breakout and SpaceInvaders, there is a large exploitation gap between $V^{\hat{\pi}^*_{D\infty}}(s_0)$ and $V^{\hat{\pi}^\theta}(s_0)$.

The practical sub-optimality may be an overestimate of true policy performance. To address this issue, we perform a pure analysis with a set of completely deterministic environments in Figure 1b, Figure 3c, Figure 8, Figure 9, and Figure 10. Because these environments are *deterministic*, it is possible to compute a true $V^{\hat{\pi}^*}(s_0)$ which is equal to the best single trajectory ever discovered. This best single trajectory is visualized as $V^{\hat{\pi}^*}(s_0)$, where the policy for $V^{\hat{\pi}^*}(s_0)$ is $a_0, \ldots, a_t$, which is replayed to visualize the score and indicate that to reach this performance, the policy $\theta$ needs to exploit this data well to reach that score. As can be seen, $V^{\hat{\pi}^*}(s_0) > V^{\hat{\pi}^*_{D\infty}}(s_0) > V^{\hat{\pi}^\theta}(s_0)$, which indicates that $V^{\hat{\pi}^*_{D\infty}}(s_0)$ may be slightly lower than the best performance, yet these trained policies struggle to produce behavior close to $V^{\hat{\pi}^*_{D\infty}}(s_0)$, indicating that often performance is limited by a lack of good exploitation.

Last, to better understand these estimators for stochastic and deterministic settings, it is important to compare deterministic vs stochastic policy performance; in this case, the stochasticity added to the policy is causing a larger difference when the policy has learned a high-value behaviour. For PPO

---

[2]In Figure 8c we instead use a deterministic environment to make the evaluation clearer

on continuous environments, this is equivalent to taking the mean of the policy, and for a discrete policy, the $\arg\max_a Q(s_t, a)$ is used. In Figure 3a the deterministic policy does poorly, this is likely because the agent quickly reaches states that are out of distribution, causing the agent to fail. Similar is true for **Montezuma's Revenge** with PPO. However, for MinAtar/Breakout and SpaceInvaders, the $\epsilon$-greedy exploration of DQN knocks the policy off high-value paths, and the deterministic policy does well, even approaching $V^{\hat{\pi}_D^*\infty}(s_0)$ for MinAtar/Breakout for PPO and PQN **??**. We also observe in Figure 1b and in many other results that the difference does not decrease with additional training, indicating that the gap is not due to needing more experience or updates, but rather to more significant changes to improve exploitation and optimization in deep learning.

## 5.2 SUB-OPTIMALITY WHEN ADDING EXPLORATION

This section asks the question *does adding exploration objectives increase the difference and therefore aggravate the exploitation challenges*. This is analyzed by adding common exploration bonuses to the RL algorithms, RND (Burda et al., 2018b). RND adds an additional reward to the extrinsic reward, encouraging the agent to explore a wider distribution of states, thereby enabling it to discover new, higher-reward states. These higher-reward states should yield larger returns, and if the algorithm is not effectively exploiting these rewards, the difference will be greater.

Figure 7 provides the results of the analysis of practical sub-optimality estimating Equation (3) compared with and without using RND. As we can see, the addition of RND improves the returns for DQN and PPO . However, the *difference* is also increased, indicating that as exploration is increased, so too are the issues of exploitation of experience in deep RL. This is an undesirable situation; as the agent improves its exploration, it actually learns less from the experience overall due to optimization issues.

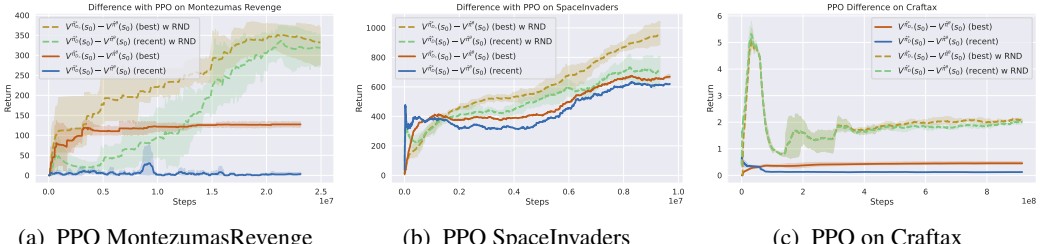

(a) PPO MontezumasRevenge     (b) PPO SpaceInvaders     (c) PPO on Craftax

Figure 4: Comparisons of practical sub-optimality for best and recent performance compared to the average using Equation (3) with and without adding **RND**. These results show that with the addition of **RND**, the difference increases, indicating that adding exploration objectives is a double-edged sword, better exploration but more difficult exploitation.

## 5.3 SUB-OPTIMALITY WHEN SCALING NETWORKS

Many recent reinforcement learning works are discovering improved algorithms' performance based on scaling networks (Schwarzer et al., 2023; Lyle et al., 2023; Obando-Ceron et al., 2024; Nauman et al., 2024; Tang and Berseth, 2024). Are the challenges from scaling just optimization issues, or are these models also struggling to scale because the types of narrow distributions produced by larger models limit exploration? Two experiments were performed to investigate this question with networks of different sizes. First across Atari environments **BattleZone** and **NameThisGame** from the Atari-5 group (Aitchison et al., 2022) that is representative of the Full Atari Benchmark, and then across **HalfCheeta**. For the Atari environments, a comparison is made between training a policy that uses the normal C-51 type network with a 3-layer CNN and using a ResNet18. For the **HalfCheetah** environment, different numbers of layers are used between 4 and 256.

In Figure 5, the results of the described experiments are given. Interestingly, the results for the Atari environments show that the difference is much larger when the policy network is a ResNet-18 instead of a 3-layer CNN. This indicates two items: one, the policy is generating higher value trajectories, but it is not adequately learning from them, and two, the gap for $V^{\hat{\pi}_D^*\infty}(s_0)$ and $V^{\hat{\pi}_D^*}(s_0)$ is very close, indicating that the policy is struggling to match these higher value experiences even when they are in the current replay buffer. With **HalfCheetah**, the issue of scale is studied by training a policy

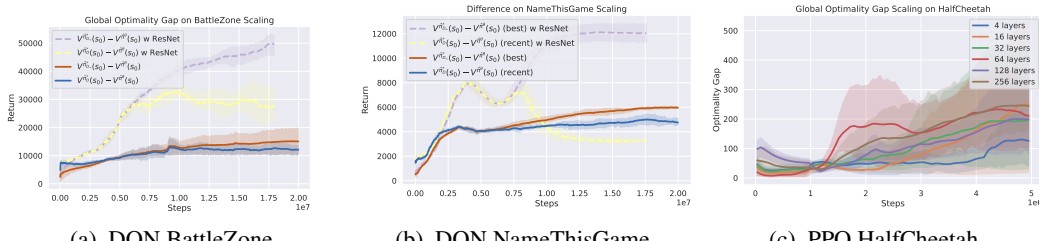

(a) DQN BattleZone      (b) DQN NameThisGame      (c) PPO HalfCheetah

Figure 5: Comparisons of practical sub-optimality for models with different-sized networks. On the left and middle, it is shown that using a ResNet-18 instead of the common 3-layer CNN for BattleZone increased the difference. On the right, the difference for **HalfCheetah** increases with the number of layers, indicating increasing exploitation issues.

over networks of 6 different sizes. In Figure 5c, the $V^{\hat{\pi}^{*}_{D\infty}}(s_0) - V^{\hat{\pi}^{\theta}}(s_0)$ is shown, and there is a trend that as the number of layers increases, the practical sub-optimality increases. This is interesting because in Figure 3a the performance with one layer is given and there is no gap. The introduction of additional layers quickly introduces exploitation issues, keeping the policy from learning the same performance in Figure 3a. This collective information suggests that scaling networks does not likely cause exploration issues, but rather reinforces the commonly understood cause of exploitation (optimization/model) issues with scale.

### 5.4 ALGORITHM SUB-OPTIMALITY

Is algorithm progress limited by weaknesses in exploration or exploitation? This question can be estimated by using the practical sub-optimality to compare aggregate analysis across tasks and RL algorithms, as described in Section 4.2. Starting with aggregate analysis across the *AtariFive* environments, we can see in Figure 6a that DQN and PPO are only able to achieve a little over 30% of the performance of their best experience (lower is better). This high value indicates that both of these algorithms struggle to produce the best possible results they have experienced. In this case, $V^{\hat{\pi}^{*}_{D\infty}}(s_0)$ (Figure 6a) is similar to $V^{\hat{\pi}^{*}_{D}}(s_0)$ (Figure 6b), indicating that the RL algorithms are experiencing high returns regularly, with a value of 0.68, they are not sufficiently capturing.

Interestingly and conversely, the *rliable* optimality gap indicates that DQN is better than PPO in Figure 6c, because DQN does achieve higher average policy performance, but the analysis from comparing to $V^{\hat{\pi}^{*}_{D}}(s_0)$, in Figure 6b shows us that even though DQN performs better than PPO, DQN is still generating a lot of high-value experience that it is not able to exploit. Conversely, because PPO is performing worse according to *rliable*, but has a better $V^{\hat{\pi}^{*}_{D}}(s_0) - V^{\hat{\pi}^{\theta}}(s_0)$, improved exploration would improve PPO more than it would DQN. Overall, these results suggest that both algorithms struggle to extract the most from their experience, and that more information can be used to compare algorithms beyond *rliable*.

## 6 DISCUSSION

This work has introduced a method to study the limitations of deep RL algorithms in the space of exploration and optimization challenges. An estimator is introduced to support this position. The estimator is used to show that common RL algorithms struggle to exploit their experience and that adding exploration bonuses and scaling networks exacerbates these issues. This estimator can be used to assist users in understanding if poor performance in an environment is the result of limited exploration (data problem) or more stable optimization to make progress (model problem). Because RL agents collect vastly different data during training, it can be difficult to compare performance across algorithms. This estimator adjusts the comparison to show how well the algorithm did compared to the distribution of collected data (experience). Because the estimator comparison over generated experience measures the sub-optimality relative to the agent's generated experience, it can be better suited to task-independent comparisons. In the future, this metric can be used to evaluate broadly across produced algorithms to assist researchers and practitioners in their analysis.

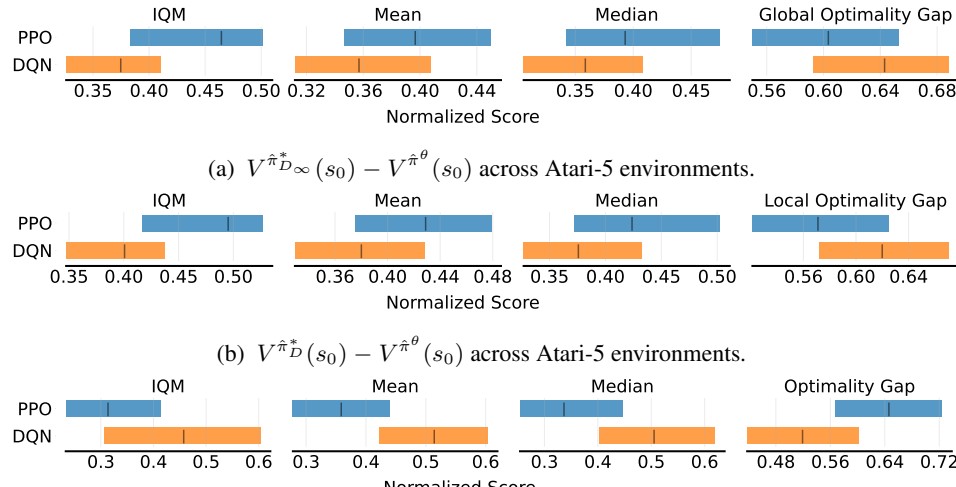

(a) $V^{\hat{\pi}^*_D\infty}(s_0) - V^{\hat{\pi}^\theta}(s_0)$ across Atari-5 environments.

(b) $V^{\hat{\pi}^*_D}(s_0) - V^{\hat{\pi}^\theta}(s_0)$ across Atari-5 environments.

(c) Normal *rliable* evaluation across Atari-5 environment.

Figure 6: *rliable* plots for **PPO** and **DQN** over AtariFive environments. This measure gives an aggregate view for each algorithm as each sample is normalized using the practical sub-optimality from each run's generated data.

**Reproducibility Statement.** We provide implementation details in the main paper. However the overall method is easier to reimpliment in the cleanrl codebase. We plan to openly release our code upon the publication of our work.

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

# A ABLATIONS

## A.1 TOP % SENSITIVITY ANALYSIS

.

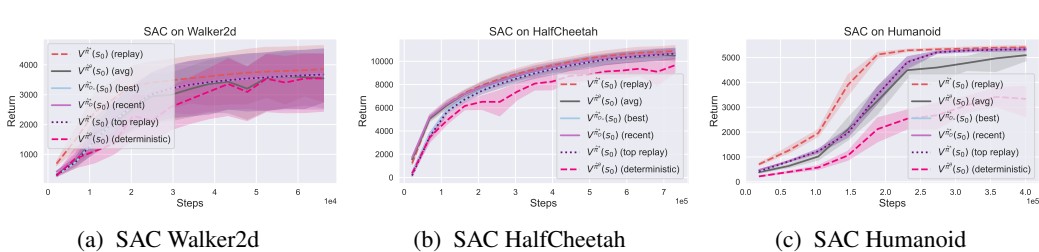

(a) PPO SpaceInvaders

Figure 7: Comparisons of practical sub-optimality over different settings for the using the top x % to compute $V^{\hat{\pi}^*}(s_0)$ in comparision to the best replay trajectory. We can see that the $V^{\pi^*}(s_0)$ which is approximiated by replaying the highest value trajectory ever is noisy and using the top 5% of recent data produces a fair trade-off in estimation but is closer to the expert behavior.

# B SAC

This section includes results on SAC (Haarnoja et al., 2018).

## B.1 RESULTS ON CONTINUOUS CONTROL ENVIRONMENTS

.

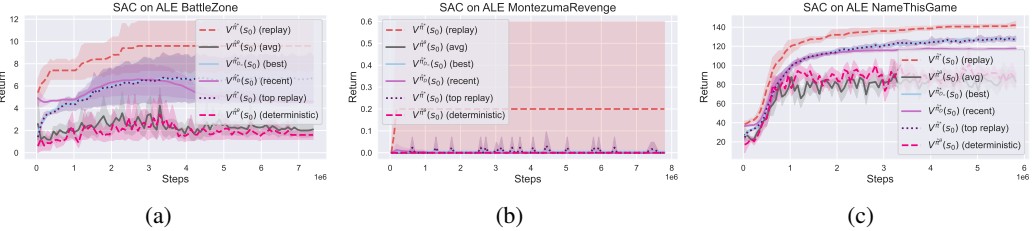

(a) SAC Walker2d          (b) SAC HalfCheetah          (c) SAC Humanoid

Figure 8: Comparisons of practical sub-optimality over SAC for continuous control environments with 10 seeds.

As we see in these results, SAC has a smaller gap between the generated data. However, the gap is increased for the humanoid environment, which is the environment with the largest action dimensionality. As SAC is designed to be better at optimization and finding an optimal policy, the results here correlate with our metric in that the gap is smaller for this RL algorithm that is designed to learn from data better.

## B.2 DISCRETE CONTROL ENVIRONMENTS

(a)          (b)          (c)

Figure 9: Comparisons of practical sub-optimality over SAC for discrete control environments in ALE with 10 seeds.

## C PQN

This section includes results on PQN (Gallici et al., 2025).

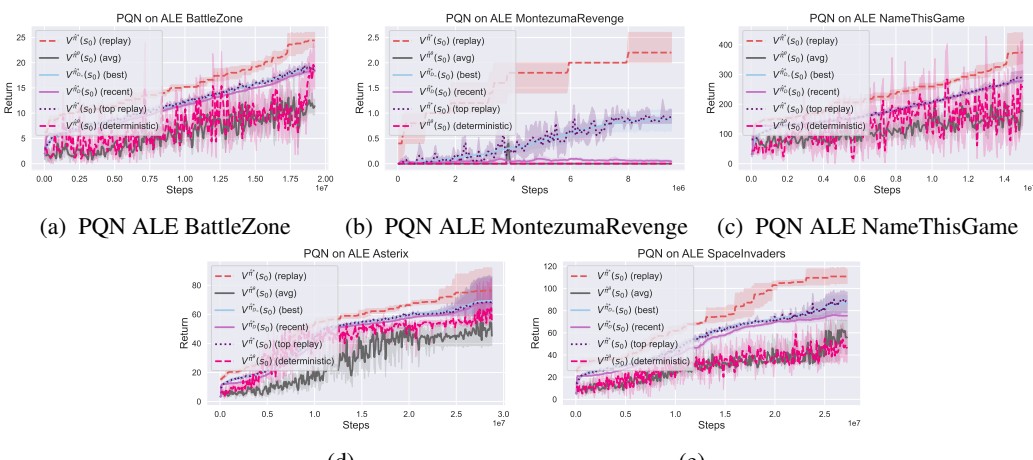

Figure 10: Comparisons of practical sub-optimality over PQN for discrete control environments in ALE with 10 seeds.

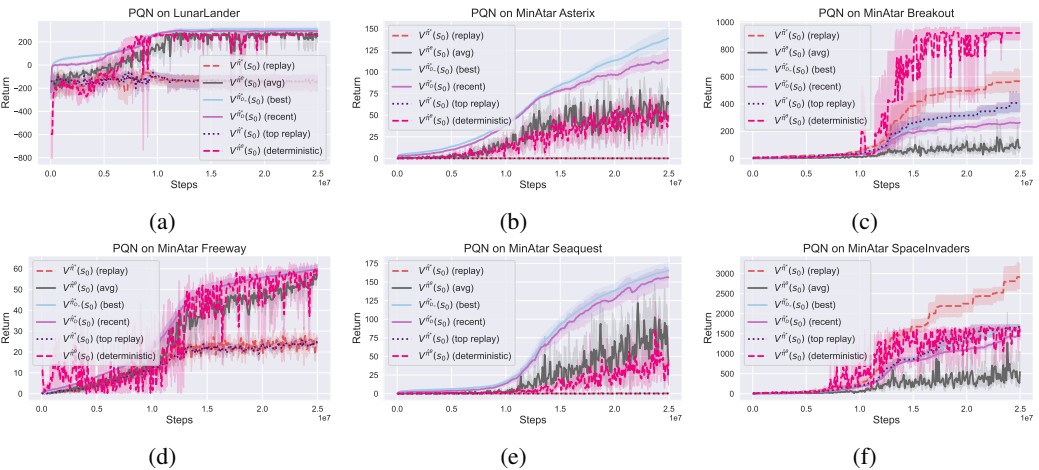

Figure 11: Comparisons of practical sub-optimality over PQN for discrete control environments with 10 seeds.

In these experiments we have added $V^{\hat{\pi}^*}(s_0)$ (top replay) which replays a collection of the trajectories from the $V^{\hat{\pi}^*}(s_0)$ distribution. We show these replays to indicate that for deterministic environments not only is the average policy far from the best trajectory but it is also far the many possible trajectories that achieve higher value than the average (avg) policy. We see this is true for Atari environments and for some of the MinAtar environment that are deterministic (Breakout and SpaceInvaders).

