# OpenReview forum: "Is Exploration or Optimization the Problem for Deep Reinforcement Learning?"
_ICLR.cc/2026/Conference — Submitted to ICLR 2026_

### Official Review · Reviewer_uSzU · 2025-10-21

**Soundness:** 1
**Presentation:** 2
**Contribution:** 2
**Rating:** 2
**Confidence:** 3

**Summary:**

This work proposes an estimator for assessing the sub-optimality and optimization limits of deep reinforcement learning algorithms. The authors show experimental results across different environments and RL methods and reveal that the proposed best experiences generated are 2–3 times better than the learned policy performance, indicating that current deep RL approaches may only exploit only about half of the high-quality experiences they produce.

**Strengths:**

* I think the potential contributions are strong, and I enjoyed the introduction of the paper, the problem is well motivated.
* Deciding if the model is failing due to exploration or exploitation is potentially a valuable insight for designing a better algorithm.
* Finding a way to deduce if a model is not exploiting high value transitions well enough would be very useful in the field of RL.
* The experiment setting spans a diverse setting of environments, showing potential transferability.

**Weaknesses:**

Unfortunately, I found this paper very difficult to read due to the lack of rigor, lack of claim backing, and missing terms. Often times, it isn't clear on what is trying to be said. For example:
* Most empirical works use $V(π_θ(s_0))$ for comparing across algorithms to understand which algorithm
performs the best on a set of tasks.
  - Can there be citations provided here?
* However, if the policy π can not explore optimally, using $π^∗$ is not very informative.
  - This wasn't clear to me. Informative with respect to exploration? Informative with respect to debugging purposes? Can this be written out explicitly?
* The challenge is that $π_θ$ can be arbitrarily bad compared to $π^∗$
  - Bad in what sense? Sub optimality?
*  If the experience were equal, algorithm B would see the same experience as algorithm A, then B would result in better
performance and have a smaller practical sub-optimality.
   - Can this be proven? This statement does not come across as trivial. If the generated experience of B was as good as A, then it would no longer be B since B is said to generate lesser experience than A. As it reads, the above statement might imply that if A was as good as exploiting as B, then A would be better? Then it is no longer A.
*  $\pi_θ$, $\hat{\pi}^*$ are broadly defined, but its not clear what $\hat{\pi}^\theta$ means, and it is never defined despite being used in section 4.
* Can more explanation and set up be shown for EQ 6?
*  For stochastic environments, the first version the best stocastic policy from the
collected experience as top 5% of experience generated by the agent $V^{∗πD_{∞}}(s_0)$, where $D_∞$is all the experience collected by the agent.
   - I feel like an appendix should be used to address the terms that are being used, what is top 5%? top 5% most common? top 5% with respect to what? Why is this considered to be the 'best' is it because it comes from the optimal policy on all data? How was that policy obtained? At this point, it is still unclear to me why EQ 6 is a suitable score to determine sub optimality.
* I feel as though since EQ 6 depends on how V is computed, it should be clearer on how the $D_∞$ is explicitly used to estimate V, assuming V is a functional approximation, the batching of the data and the converge should be talked about to inform the reader about the precautions taken.
* Figure 1 states expert policy, to better align with the rest of the writing, should this be changed to the 'optimal policy'?
* After training DQN on Montezuma’s Revenge (Figure 3b)
  - PPO?

There seems to be a lot of terms used but no trace to what they explicitly mean. Claims seem to be stated, and quickly moved on from without any explanation or proof. This paper would greatly improve if the authors utilized an appendix, began to prove some of the claims used and were clear on the definitions used. Due to the clouded structure I unfortunately found it difficult to digest the insights made in the experiment section and compare with findings in current RL literature.

**Questions:**

Besides popularity as a choice of agent, have the authors considered actor-critic agents? Being that V is computed, it would be interesting to compare how this is different to learning the critic in Soft Actor Critic, when the agent is actively updating its own value function.

---

> ### Author Response · Authors · 2025-12-02
> **Authors Updates**
>
> Thank you for your feedback and comments on the strong contribution of the work. Below are comments to improve clarity and further increase the impact with additional experiments and definitions.
>
> Q1: ICLR 2025 reviews deeprl sub-opt Lack of clarity
>
> A1: Thank you for pointing out this lack of clarity. We have gone over the text given the feedback and updated many sections of the paper to be more clear and to the point on the terms and definitions.
>
> Q2: ICLR 2025 reviews deeprl sub-opt: Argument that algorithms that learn better from their data are stronger RL algorithms.
>
> A2: The paper proposes that algorithms that have a smaller practice exploitation gap are better RL algorithms. The paper introduces an example were is two algorithms generate the same data, the one that is able to better exploit the higher value trajectories, to achieve higher value for the learning policy is the better algorithm. This also implies that if two algorithms achieved the same value for their learned policy the one with lower value experience optimal policy is the better algorithm, because it did not waste time on exploring data it can not learn from. The wording around this concept has been updated in the paper to make this more clear.
>
> Q3: ICLR 2025 reviews deeprl sub-opt Can more explanation and set up be shown for EQ 6?
>
> A3: Yes, this can be connected more clearly to rliable [1] to show what is being explained. In the rliable paper the authors propose an improved statistical analysis to compare algorithms. That comparison relies on an estimation of the optimal policy $V(s)^{\pi^{\theta}}/V(s)^{\pi^{*}}$, for which we have well known data points for the human level performance on Atari, but we do not have those values in general. Our work proposes that we can easily use the experience optimal policy in place of the expert policy value to help evaluate policy performance. By using the normalization it is easier to compare algorithms and groups results together statistically. rliable will then compare the IQM of the results across tasks $\mathcal{T}$. We use the same statistical analysis but replace the human normalized scores with our experience optimal policy estimator. We have updated the paper to make this more clear, and how the method connects to rliable.
>
> Q4: How is V learned
>
> A4: V is not learned from data in our work, it is the average return of experience generated by the agent $D$.
>
> Q5: How is the top 5% computed?
>
> Q5: If we sort the experience with the highest value trajectories at index 0, then the top 5% is the first 5% of trajectories in that sorted dataset. Essentially the to 5% (now the top k%, in the paper to indicate that we can alter change this number), is the top x % of experience the learning policy has generated, we use the sorted example here to indicate the process.
>
> [1] Agarwal, Rishabh, Max Schwarzer, P. S. Castro, Aaron C. Courville, and Marc G. Bellemare. 2021. “Deep Reinforcement Learning at the Edge of the Statistical Precipice.” Edited by M. Ranzato, A. Beygelzimer, Y. Dauphin, P. S. Liang, and J. Wortman Vaughan. Neural Information Processing Systems 34 (August): 29304–20.

---

### Official Review · Reviewer_LnpL · 2025-11-01

**Soundness:** 2
**Presentation:** 2
**Contribution:** 2
**Rating:** 4
**Confidence:** 3

**Summary:**

This paper empirically studies the performance of an RL algorithm by separating its ability to explore from its ability to exploit. Exploration is defined here as the ability to acquire data, and exploitation as the ability to learn the best possible policy (optimization problem) from the acquired data. The study is done by introdcuing a measure of the best policy that the algorithm can learn from the acquired data to measure the "exploitation" capability.

**Strengths:**

1. The paper covers an important topic in RL.
2. The proposed definition of exploitation is interesting and allows us to study the convergence of algorithms in a different way.
3. The experimental results are extensive.

**Weaknesses:**

Here are the main weaknesses I have identified, some of which can be considered issues that should be clarified.
1. The literature review focuses mainly on problems associated with value-based methods. However, the paper claims to study convergence in general. Typically, for policy gradients, numerous papers have studied and proven convergence towards (global) optima. These often have a hidden assumption of necessary exploration. These results are not discussed.
2. I feel that the paper overlooks the fact that the exploration method influences the convergence of algorithms in practice. Typically, in the case of policy gradient, entropy regularization makes it possible to eliminate local optima, make optimization robust, etc. This literature is not discussed in the paper.
3. The literature on exploration methods is rather sparse; it does not cover maximum entropy methods, for example.
4. In general, the paper is quite difficult to follow. E.g., the definition used for exploration/exploitation is given quite late in the text. The text attempts to provide insight, which is obviously good, but I feel that this sometimes obscures the exactness of certain definitions/discussions.
5. From what I understand, the paper deals with a fairly general problem of learning policy from a set of data. Here referred to as exploitation. I get the impression that this is a fairly generic and well-studied problem. Convergence of methods, global optima, overfitting/underfitting, etc. Yet the problem is presented as stand-alone here.

Some (non-extensive) elements of the literature may be of interest concerning my previous remarks. I, of course, understand that it is impossible to cite everything and that finding the most related studies is important.
Concerning convergence:
1. Bhandari, J. and Russo, D. Global optimality guarantees for policy gradient methods.
2. Bhatt, S., Koppel, A., and Krishnamurthy, V. Policy gradient using weak derivatives for reinforcement learning.
3. Agarwal, A., Kakade, S. M., Lee, J. D., and Mahajan, G. Optimality and approximation with policy gradient methods in markov decision processes.
4. Zhang, J., Kim, J., O’Donoghue, B., and Boyd, S. (2021a). Sample efficient reinforcement learning with reinforce.
5. Montenegroa, A., Cesania, L., Mussia, M., Papinia, M., and Metellia, A. M. Learning Deterministic Policies with Policy Gradients in Constrained Markov Decision Processes.

Papers studing influence of exploration on PG:
1. Husain, H., Ciosek, K., and Tomioka, R. Regularized policies are reward robust.
2. Ahmed, Z., Le Roux, N., Norouzi, M., and Schuurmans, D. Understanding the impact of entropy on policy optimization.
3. Bolland, A., Lambrechts, G., & Ernst, D. Behind the myth of exploration in policy gradients.

**Questions:**

1. Could you clarify if the elements from the literature previously highlighted are related to the current study?
2. The study is largely justified by the problems caused by non-IID transitions. Your method does nevertheless not study this problem explicitly, from my understanding. Other phenomena, such as the existence of local optima, convergence speed of local search methods, and overfitting, could be the cause of the inefficiency too. Am I right, and would these phenomena be captured by your metric? Also, in PG there is no real problem of IID transitions, as algorithms consider full trajectories that are IID. In PPO, for example, what does the metric tell us about the influence of non-IID samples on the algorithm?
3. I did not understand equation (4) and section 4.1 beyond intuition. Typically, how does policy (4) choose actions knowing that not all states are in $D^\infty$? Similarly, I find equation (5) counterintuitive; I interpret it as a redefinition of the value function. Can the authors clarify the equations?
4. In my opinion, the claims are a little strong, given that the metric has no theoretical basis and is influenced by several factors. Do we really show that the current problem in RL lies in the ability of algorithms to exploit the information acquired?

---

> ### Author Response · Authors · 2025-11-26
> **Author Updates**
>
> A1: ICLR 2025 reviews deeprl sub-opt Connections to convergence guarantees for algorithm
>
> Q1: The majority of these convergence guarantees in prior work focus on the use of linear models, and this work's primary focus is the combination of RL objectives with non-linear deep models, where convergence guarantees are not clear. We agree that prior work on convergence algorithms is important, and our work aims to understand the gap between RL objectives and their interplay within deep learning models.
>
> A2: Connections on how exploration effects convergence of the algorithms
>
> Q2: The effects of exploration are a main point in the paper. This point is that just adding intrinsic rewards or other exploration methods or RL algorithm is non-trivial. Those RL algorithms have their distributions that enable their convergence. Our results in Figure 4, where we add RND to common RL algorithms, show that the optimization gap increases, thereby widening the performance gap between the generated data and the learned policy.
>
> Q3: Maximum entropy methods are missing
>
> A3: Entropy regularization is indeed designed to help with the above issues so we have added experiments using SAC, however, we see that even SAC struggles to close the optimization gap. This gap is particularly large for the Atari environments and the humanoid environment, where the number of action dimensions is large.
>
> Q4: Link to prior discussion on convergence
>
> A4: We appreciate that there is prior work on the convergence of reinforcement learning algorithms under assumptions that the model is linear. This work is specifically in the space of deep learning and reinforcement learning where many of these convergence findings do not directly apply. Also, deep reinforcement learning is being used more widely in and outside the research community, making it more important to help provide tools for the community to understand the issues of RL algorithms.
>
> Q5: Clarity and definitions
>
> A5: We agree and have improved the definitions and description in the text. There were also some errors in the notation that may have led to this confusion

---

### Official Review · Reviewer_ajJV · 2025-11-07

**Soundness:** 2
**Presentation:** 2
**Contribution:** 2
**Rating:** 4
**Confidence:** 4

**Summary:**

This paper aims to study an important problem: for practical deep reinforcement learning problems, is exploration or optimization (exploitation) a more severe problem? This paper has discussed ideas to analyze this problem in Section 4, and experimental results are demonstrated in Section 5.

**Strengths:**

- The studied problem, "is exploration or optimization (exploitation) the problem for (practical) deep RL", is an important and interesting problem.

- The experimental results in Section 5 are interesting.

**Weaknesses:**

Though this paper is interesting, I do not think the current version is ready for publication, for the following reasons:

- [major] The presentation and discussion in Section 4 are too handwavy. I recommend that the authors make a major revision of it to make it more rigorous. Most importantly, please provide a mathematically rigorous definition of the **experience optimal policy** $\hat{\pi}^*$. This is crucial, since the experience optimal policy is a key concept in this paper, and is used to identify whether exploration or optimization is the problem.

Section 4 has also provided methods to compute or estimate the experience optimal policy or its value. With a rigorous definition, this paper should also discuss whether these computes or estimates are accurate, and if not, how large the estimation errors are.

- This paper has many inconsistent and undefined notations, for instance:
  - the reward function is defined as $R(s_t, a_t)$ in Section 3, but $r(a_t, s_t)$ in Section 4
  - in Section 4, both $\pi^\theta$ and $\hat{\pi}^\theta$ are used for "achieved exploitation"
  - Please provide rigorous definitions for $V^{\hat{\pi}^\ast_{D_\infty}}$ and $V^{\hat{\pi}^\ast_{D}}$ in Section 4

Please double-check the notations and fix the typos.

- [Minor] Please make the figures in this paper larger so they are more readable.

**Questions:**

Please address the weaknesses listed above, especially the first one.

---

> ### Author Response · Authors · 2025-12-02
> **Author comments.**
>
> Thank you for your interest in the paper and comment on the problems importance and the interest in the experimental results.
>
> Q1: ICLR 2025 reviews deeprl sub-opt More formal definition of experience optimal policy
>
> A1: We have improved the clarity for the definition of the experience optimal policy. We note that the definitions of simple and essentially, the experience optimal policy is the highest value trajectory generated from the learning agent. However, this can have high variance so instead we introduce an estimator that takes into consideration the top $n \\% $ of the data. A comparison to this __softer__ estimator is a more fair evaluation of the practice exploitation gap. We have updated the paper to make this notation and description clearer.
>
> Q2: Notation updates
>
> A2: Thank you for the notes on the notation discrepancies, those have been correct. IN addition, we improve notation elsewhere in the paper to make the style more consistent.

---

### Official Review · Reviewer_Ys9d · 2025-11-08

**Soundness:** 1
**Presentation:** 3
**Contribution:** 2
**Rating:** 2
**Confidence:** 4

**Summary:**

The paper’s core claim is that modern deep RL struggles less with finding rewarding behavior and more with turning those good experiences into strong policies. To tease this apart, the authors propose a simple “practical sub-optimality” estimator: compare a learned policy to the best trajectories it has already produced (the “experience-optimal policy”). Across PPO and DQN on Atari/MinAtar, MuJoCo, Montezuma’s Revenge, and Craftax, they observe large gaps—often 2–3×—between those best trajectories and the final policy. They also find the gap tends to grow when you add exploration bonuses like RND or scale up the model, reinforcing that the bottleneck is exploitation/optimization rather than exploration.

**Strengths:**

The paper is generally clear and well-structured. It is well-motivated and takes on an important, under-discussed question: when deep RL stalls, is the bottleneck exploration or exploitation? Bringing this issue to the forefront is valuable for both researchers and practitioners and, in my view, warrants attention regardless of whether one agrees with the specific estimator proposed.

**Weaknesses:**

* Soundness of the estimator: Using the top 5% highest-return trajectories as a proxy for the “experience-optimal policy” is problematic in stochastic environments. High-return episodes may result from lucky transitions or risky, low-expectation action sequences, making them non-reproducible and not necessarily exploitable by a learned policy.
* Lack of analysis on learnability: The paper does not examine whether these “good trajectories” correspond to behavior that is actually learnable or generalizable. Without connecting high-return episodes to stable, reproducible structure, it is unclear whether the measured gap truly reflects an exploitation failure rather than noise.
* Arbitrary percentile choice: The decision to use the top 5% of trajectories is not theoretically or empirically justified. The paper does not discuss sensitivity to this choice or explain why this particular percentile should meaningfully approximate an experience-optimal policy.
* Insufficient number of seeds: All experiments use only 4 seeds, which is too few to support strong conclusions

Overall, the soundness of the proposed estimator remains unclear, and I feel the paper overclaims by asserting that exploitation is the primary bottleneck in deep RL; the experiments and reasoning do not sufficiently support such a broad conclusion

**Questions:**

The paper repeatedly uses $V(s\_0)$ in discussion, but since most environments sample $s_0$ from an initial-state distribution, shouldn’t the evaluation be expressed as an expectation $\mathbb{E}_{s_0}[V(s_0)]$ rather than a single-state value?

---

> ### Author Response · Authors · 2025-11-26
> **Author Updates**
>
> Q1: ICLR 2025 reviews deeprl sub-opt: Using the top 5% highest-return trajectories as a proxy for the “experience-optimal policy” is problematic in stochastic environments. High-return episodes may result from lucky transitions or risky, low-expectation action sequences, making them non-reproducible and not necessarily exploitable by a learned policy.
>
> A1: It is true that in stochastic environments, it may be difficult to reproduce the return for good trajectories. This is why we include experiments on discrete environments as well, where we can see in Figure 1b,3c that the average policy is much worse than the replayed trajectory, indicating a large exploitation gap. In addition, we have now added many more experiments to the appendix for SAC and PQN on more deterministic environments (Atari, MinAtar, Mujoco). The gap between the average policy and the replayed trajectories is growing for most environments. This indicates that many deep RL algorithms struggle to learn from their own data.
>
> Q2: ICLR 2025 reviews deeprl sub-opt it is unclear whether the measured gap truly reflects an exploitation failure rather than noise.
>
> A2: Continuing the discussion from (A1) in the controlled experiments in 1b and 3c where the environment is completely deterministic, because we created a custom version of MinAtar without noise, we can show without a doubt that there is a large difference between the (replay), which replays a fixed sequence of actions in the environment, and the learned (avg) policy. This difference is the optimization gap that measures how much an RL algorithm “leaves on the table” by not learning from its own experience. We do not extend these results to additional RL algorithms (SAC and PWN) in the Appendix, where we include even more experiments across additional environments where we carefully make the conditions deterministic.
>
> Q3: ICLR 2025 reviews deeprl sub-opt: The decision to use the top 5% of trajectories is not theoretically or empirically justified.
>
> A3: Good point, this deserves further analysis. We have performed analysis over a number of different settings for this percentage and put the results in the appendix. We can see there is a very high variance over the estimation as we reduce the percentage below 5% and as we include more above 5% the estimate looks more like the average policy. From this 5% still look like a fair trade-off between estimating a reasonable upper performance that captures that there is not just one good trajectory that the agent can not match, but many.
>
> Q4: Insufficient number of seeds
>
> A4: We are working on increasing the number of seeds to 10. The original experiments took a considerable amount of compute across algorithms, environments, and configurations with RND/ResNet/DeepNetworks. The additional experiments will require 80 GPUs running concurrently for a week. We have focused these efforts on adding new rl algorithms, SAC and PQN. We have added these new experiments to the appendix. The inclusion of these new algorithms has not changed the findings from the original paper. SAC and PQN both struggle to learn from their data.

---

### Official Review · Reviewer_VZUT · 2025-11-11

**Soundness:** 3
**Presentation:** 1
**Contribution:** 3
**Rating:** 6
**Confidence:** 3

**Summary:**

This paper investigates a central and long-standing question in Deep Reinforcement Learning (DeepRL): Is the bottleneck in current deep RL performance primarily due to insufficient exploration or to optimization inefficiencies? To answer this, the authors propose a novel “practical sub-optimality estimator” that measures the gap between: The best experience ever collected by the agent (the experience-optimal policy, denoted \pi^*_D), and The learned policy’s performance V_{\pi_\theta}. If the difference between V_{\pi^*D} and V{\pi_\theta} is large, it implies an exploitation/optimization problem; if the gap is small, it suggests the main issue is exploration.
Using this estimator, the paper analyzes common algorithms such as PPO and DQN across a diverse set of environments, including Atari, MinAtar, Craftax, and MuJoCo (HalfCheetah, Walker2d, Humanoid), and under variations such as adding RND exploration bonuses and scaling network size (from small CNNs to ResNet-18). Key findings include 1) The gap between best experience and learned policy performance is often 2–3×, implying that deep RL methods only exploit roughly half of the good experiences they generate. 2) Adding exploration bonuses (e.g., RND) increases the gap, showing that better exploration can worsen optimization inefficiency. 3) Scaling network size also increases sub-optimality, reinforcing the claim that optimization and exploitation, not exploration, are the main limitations of current deep RL methods.

**Strengths:**

- Clear and well-motivated research question. The paper addresses a fundamental question in RL—exploration vs. exploitation—that has often been debated but rarely quantified. The proposed estimator provides a concrete diagnostic tool to analyze this trade-off empirically.
- Broad experimental coverage. The authors test across diverse environments and both on-policy and off-policy algorithms (PPO and DQN). Figures 3–5 (pp. 6–8) show consistent trends across MinAtar, Atari, Montezuma’s Revenge, HalfCheetah, etc., supporting the generality of conclusions.
- Potentially useful diagnostic tool. The proposed sub-optimality metric could serve practitioners as a diagnostic to quickly determine whether performance limits stem from poor exploration or optimization, guiding algorithmic focus.

**Weaknesses:**

- The proposed metric compares best vs. average trajectories but does not causally separate exploration and optimization. For example, an algorithm’s “best experience” may depend heavily on stochastic exploration artifacts rather than a genuine ability to generate diverse high-value data.
- Limited algorithmic diversity. The experiments focus mainly on PPO and DQN, which, while standard, represent only a subset of deep RL paradigms. Missing are modern algorithms such as SAC, IQL, which emphasize optimization stability and could challenge the generality of conclusions.
- The conclusion that “deep RL is mainly limited by exploitation” might overstate the findings. In complex domains (e.g., sparse reward tasks like Montezuma’s Revenge), exploration remains a fundamental challenge, even if optimization inefficiency also plays a role.
- The observed “exploitation gap” could arise from various factors—bootstrapping noise, representation drift, catastrophic forgetting—not purely optimization failure. A deeper analysis separating these sources would strengthen the claim.
- While the paper is rich in figures (e.g., Fig. 3–6), it lacks aggregated quantitative summaries (e.g., average sub-optimality across tasks in tabular form). This would help communicate results more clearly.

**Questions:**

N/A

---

> ### Author Response · Authors · 2025-11-26
> **Author Updates**
>
> Q1 Limited algorithmic diversity. ICLR 2025 reviews deeprl sub-opt
>
> A1: The paper includes PPO and DQN, which are the most popular algorithms in use; however, we agree that adding additional algorithms will expand the analysis. In addition we have now added results for SAC and PQN to the appendix. SAC will also address issues in comparison to algorithms that are based on maximum entropy optimization, which may result in finding more optimal policies. The results on SAC and PQN support our conclusions that many RL algorithms struggle to learn from their generated data. While SAC has a smaller gap for tasks with fewer actions in continuous control, it does not perform well with higher action dimensions or the discrete atari environment,s where the deep networks are more complicated.
>
> Q2: ICLR 2025 reviews deeprl sub-opt In complex domains (e.g., sparse reward tasks like Montezuma’s Revenge), exploration remains a fundamental challenge, even if optimization inefficiency also plays a role.
>
> A2: While it is true that exploration is challenging in Montezuma’s Revenge there is also a significant deficiency in learning from the generated data. Looking at Figure 3.b in the paper, we can see that PPO regularly generates much higher-quality data than the average policy performance, about 4x better. The difference between these two signifies a gap that is larger than the difference between the average policy and the initial randomized policy, indicating that in this case, optimization is a larger cause of the gap in performance. It is this point exactly that this work focuses on: that generating good data is not enough, and that without additional considerations for the exploitation challenges, new data exploration algorithms may appear worse than they actually are.
>
> Q3: The observed “exploitation gap” could arise from various factors—bootstrapping noise, representation drift, catastrophic forgetting
>
> A3: There appears to be some confusion here in the use of the term optimization, by optimization, we do not mean only the gradient update process but the many mechanisms that make deep learning difficult, including bootstrapping noise, representation drift, forgetting, etc, which all limit the deep learning model from learning from the generated data. We have updated the terminology used in the paper to make this clearer. We agree deep analysis of each factor for each algorithm can add additional insight as to the causes of learning little from he generated data; however, we leave that to future work.
>
> Q4: ICLR 2025 reviews deeprl sub-opt:  rich in figures (e.g., Fig. 3–6), it lacks aggregated quantitative summaries
>
> A4: The paper does not include a table of these aggregate metrics, but this aggregate analysis is included in Figure 6. Figure 6 aggregates the optimization gap across all environments used in the paper for each algorithm, PPO and DQN. In fact, the new analysis in the paper, using the optimality gap metric that compares performance relative to the generated data, indicates that PPO is good at learning from its data, better than DQN, which is different from the standard reliable metric that considers algorithm performance independent of its own distribution.

---

### Author Response · Authors · 2025-11-26
**General response**

We thank the reviewers for their feedback on the work.

Our work focuses on the challenge of training deep learning models in reinforcement learning environments. Many reviewers have described the importance of having such a tool will have a large impact in the community (VZUT, VZUT11, Ys9d, LnpL, uSzU21, All reviewers). This work offers a new method and metric for evaluating the performance of different RL algorithms, helping practitioners understand whether poor performance is due to poor exploration or poor exploitation of the generated data. The paper proposes to measure poor exploitation by looking at the difference between the best data the agent generates compared to its average performance. Using this metric, we run experiments with many popular RL algorithms across common environments and find that, in many cases, RL algorithms generate very high-value trajectories, but the average performance is far from this level. We also include analysis of deterministic environments to clearly indicate that the agent just needs to learn to copy the high-value trajectories to perform well, but in many cases can not. Given that RL is becoming more popular, information and tools like these will be widely used to help researchers and practitioners improve their algorithms.


Summary of updates to address reviewer concerns.
Q1: More experiments to evaluate other RL algorithms (VZUT11, Ys9d)

A1: We have now included additional RL algorithms to better understand how the exploitation gaps effects a wider distribution of RL algorithms popular in the community. We have not included experiments with PQN and SAC across many environments to evaluate their ability to learn from their generated data. We had that these algorithms also indicate a exploitation gap, especially for more difficult environments. Overall, we have included about 3 times as many experiments in the paper. Including with additional random seeds, and an ablation analysis of hyperparameters.

Q2: Clarity on findings in the paper. (VZUT11, Ys9d, ajJV)

A2: The review process helped making findings, such as the challenges in using intrinsic reward methods which can increase the exploitation gap, which indicated a difficulty in learning from the new more diverse data.

Q3: Clarity of contribution (LnpL, VZUT11)

A3: Extending from above, our contribution is a method to understand the limitation of RL algorithms in general, but also how to debug RL algorithms as they are being made. RL is becoming even more popular and used by many inside and outside research, this metric makes it easier for people to understand the challenges of deep reinforcement learning. Our work also indicates that the challenge of getting deep learning models to learn in the RL paradigm is a large issue and progress on new objectives, intrinsic rewards, or theoretical bounds can be limited by the deep learning challenges more than the RL challenges.


We have not added new experiments that include SAC and PQN to the appendix, and find that many of the same challenges arise for these algorithms as well.

---

> ### Comment · Area_Chair_jzne · 2025-11-26
> **Author-Reviewer Discussion**
>
> Dear reviewers,
>
> Please review the authors' response and adjust your rating accordingly. If you have any further questions, please discuss with the authors further.
>
> AC

---

### Meta-Review · Area_Chair_ot8T · 2026-01-10

**Summary:**

This paper argues that DRL performance is bottlenecked more by optimization than by exploration. They propose a "practical sub-optimality estimator" that compares the value of the agent's learned policy against an "experience-optimal policy" (derived from the top ~5% of trajectories generated during training). Experiments show the "best" data generated is often 2–3x better than the final learned policy, suggesting agents fail to exploit their own good experiences.

While reviewers agree the research question is important and well-motivated, the majority believe the paper lacks the necessary rigor, clarity, literature review, and theoretical soundness for acceptance. The authors' rebuttal focuses more on quantity (adding more experiments to the appendix) rather than quality (being more theoretically rigorous). The AC also felt there is a logic gap between the experiments and the conclusion: i.e., if optimization is the issue, the authors may need to demonstrate that the algorithm could actually learn to be optimal given that data (e.g., via supervised learning/behavior cloning). The AC is not convinced that the paper is at a stage to be accepted.

**Reviewer Concerns:**

Addressed:
- Algorithmic Diversity: Added SAC and PQN experiments to the appendix (addressing Reviewers VZUT, LnpL).
- Deterministic Baselines: Included experiments in deterministic environments to show the gap isn't solely due to noise (addressing Reviewer Ys9d).
- Clarity: Fixed minor notation inconsistencies (addressing Reviewers ajJV, uSzU).

Remaining Concerns: The majority of other concerns do not seem to be adequately addressed.

**Reviewer Scores:**

The AC would predict that the reviewers would not increase the rating, given the current discussions.

---

### Decision · Program_Chairs · 2026-01-26

Reject